# The Proteome of Community Living *Candida albicans* Is Differentially Modulated by the Morphologic and Structural Features of the Bacterial Cohabitants

**DOI:** 10.3390/microorganisms8101541

**Published:** 2020-10-07

**Authors:** Thuyen Truong, Li Mei Pang, Suhasini Rajan, Sarah Sze Wah Wong, Yi Man Eva Fung, Lakshman Samaranayake, Chaminda Jayampath Seneviratne

**Affiliations:** 1Oral Sciences, Faculty of Dentistry, National University of Singapore, Singapore 119085, Singapore; thuyentruong@u.nus.edu; 2National Dental Research Institute Singapore (NDRIS), National Dental Centre Singapore, Singapore 168938, Singapore; pang.li.mei@ndcs.com.sg; 3Walther Straub Institute of Pharmacology and Toxicology, Member of the German Center for Lung Research (DZL), Medical Faculty, LMU-Munich, 80336 Munich, Germany; suhasini.rajan@lrz.uni-muenchen.de; 4Molecular Mycology Unit, Institut Pasteur, CNRS, UMR2000, 10098 Paris, France; sarah.wong@pasteur.fr; 5Department of Chemistry and State Key Laboratory of Synthetic Chemistry, The University of Hong Kong, Pokfulam Road, Hong Kong, China; eva.fungym@hku.hk; 6College of Dental Medicine, University of Sharjah, Sharjah 27272, UAE; lsamaranayake@sharjah.ac.ae; 7Faculty of Dentistry, The University of Hong Kong, Hong Kong, China; 8Oral Health Academic Clinical Programme, Duke-NUS Medical School, Singapore 169857, Singapore

**Keywords:** polymicrobial interkingdom biofilms, gene ontology analysis, limma differential expression analysis, cluster analysis, label free mass spectrometry analysis

## Abstract

*Candida albicans* is a commensal polymorphic and opportunistic fungus, which usually resides as a small community in the oral cavities of a majority of humans. The latter eco-system presents this yeast varied opportunities for mutualistic interactions with other cohabitant oral bacteria, that synergizes its persistence and pathogenicity. Collectively, these communities live within complex plaque biofilms which may adversely affect the oral health and increase the proclivity for oral candidiasis. The proteome of such oral biofilms with myriad interkingdom interactions are largely underexplored. Herein, we employed limma differential expression analysis, and cluster analysis to explore the proteomic interactions of *C. albicans* biofilms with nine different common oral bacterial species, *Aggregatibacter actinomycetemcomitans*, *Actinomyces naeslundii*, *Fusobacterium nucleatum*, *Enterococcus faecalis*, *Porphyromonas gingivalis*, *Streptococcus mutants*, *Streptococcus sanguinis*, *Streptococcus mitis*, and *Streptococcus sobrinus*. Interestingly, upon exposure of *C. albicans* biofilms to the foregoing heat-killed bacteria, the proteomes of the fungus associated with cellular respiration, translation, oxidoreductase activity, and ligase activity were significantly altered. Subsequent differential expression and cluster analysis revealed the subtle, yet significant alterations in the *C. albicans* proteome, particularly on exposure to bacteria with dissimilar cell morphologies, and Gram staining characteristics.

## 1. Introduction

*Candida albicans* is a commensal polymorphic constituent fungus of the healthy oral microbiome of a majority of humans [1,2]. In most individuals, this yeast resides as a harmless oral commensal but when the host immune function is impaired, it overgrows and causes varying forms of oro-pharyngeal infections [2,3,4]. *C. albicans* may also enter the blood stream causing life-threatening systemic infections with mortality rates that may reach as high as 40 percent [5]. 

*Candida* possesses remarkable ability to interact with other commensal and pathogenic bacteria to form surface attached communities termed biofilms [6,7]. Candidal-bacterial interactions are mutually beneficial, and assist the survival, persistence, and pathogenicity of the yeasts in diverse oral niches [8,9]. Indeed, mixed, interkingdom biofilms are known to display higher antifungal or antibiotic resistance as compared to mono-species biofilms [9]. Despite the clinical importance of the mixed species biofilms in the pathogenesis of various oral and systemic diseases, little is known of the social and chemical dynamics of such biofilm communities, and in particular, there is no information on the fungal-bacterial biofilm proteome.

Proteomics [10,11], the large scale study of proteomes, permits the rapid identification of sets of proteins uniquely expressed by a microbial strain under a given condition. Different computational approaches could be employed to analyze such proteomic data and determine key proteins or clusters of proteins that are significantly altered in response to a specific stimulant. One recently developed technique in this context is the differential expression analysis of the proteome based on linear modeling (limma) [12], and another is the cluster analysis based on average-linkage method of Sokal and Michener [13]. The latter is commonly used to analyze gene expression data.

In order to shed light on the mechanistic insights of how oral bacteria impact protein expression in *Candida* biofilms, we employed both the limma differential expression analysis [12] and the cluster analysis [13] methods. In particular, we evaluated the proteomic responses of *C. albicans* biofilms exposed to different heat-killed bacteria. A total of nine common oral bacteria species with structural dissimilarities were used to monitor the interkingdom interactions namely, Gram-positive bacteria: *Actinomyces naeslundii, Enterococcus faecalis, Streptococcus mutans, Streptococcus sanguinis, Streptococcus mitis*, and *Streptococcus sobrinus* and Gram-negative bacteria: *Aggregatibacter actinomycetemcomitans, Fusobacterium nucleatum,* and *Porphyromonas gingivalis*.

Interestingly, initial gene ontology analysis of the *Candida* proteomes suggested that the proteomes related to cellular respiration, translation, oxidoreductase activity and ligase activity were altered when *C. albicans* biofilms were exposed to heat-killed bacteria. Subsequent differential expression analysis revealed the key proteins and cellular pathways of the yeast that were differentially expressed by Gram-positive and -negative bacteria. In addition, cluster analysis of proteomic responses uncovered signature protein expression patterns in *C. albicans* biofilms exposed to heat-killed bacteria. Taken together, these results suggest that bacteria with dissimilar cell morphologies, and Gram staining characteristics modulate the proteomic expressions of the yeast.

## 2. Materials and Methods

### 2.1. Biofilm Formation

Biofilm development protocol was adapted from our previous publications [14,15]. A standard laboratory *Candida albicans* strain, SC5314, was used for the experiments. *Candida albicans* was cultured in YPD (1% yeast extract (BD Biosciences, Singapore, catalog number: 212750), 2% peptone (BD Biosciences, Singapore, catalog number: 211677), and 2% glucose (Sigma-Aldrich, Singapore, catalog number: G8270)) or Glucose Minimal Media (GMM) (0.679% yeast nitrogen base (BD Biosciences, Singapore, catalog number: 291940 and 2% glucose (Sigma-Aldrich, Singapore, catalog number: G8270)) and incubated at 37 °C for 18 h. The cultures were then used to prepare fungal suspensions of MacFarland of 0.38 (equivalent to 10^7^ cell/mL). An aliquot of 100 µL of the suspension was then inoculated into each well of a sterile flat-bottomed 96-well plate (Greiner Bio-One, Frickenhausen, Germany, catalog number: 655160) and incubated at 37 °C, shaking at 80 rpm for 1.5 h. Media containing the non-adhered cells were removed and washed twice with 150 µL of Phosphate-Buffered Saline (Gibco, Grand Island, NY, USA). Adhered cells were refreshed with 200 µL of GMM. The plate was incubated at 37 °C for 48 h. Media was changed every 24 h.

### 2.2. Bacterial Culture Conditions

Nine common oral bacteria species, Aggregatibacter actinomycetemcomitans (ATCC 43718), Actinomyces naeslundii (ATCC 12104), Fusobacterium nucleatum (ATCC 25586), Enterococcus faecalis (ATCC 29212), Porphyromonas gingivalis (ATCC 33277), Streptococcus mutans (ATCC 35668), Streptococcus sanguinis (ATCC 10556), Streptococcus mitis (ATCC 15914), and Streptococcus sobrinus (ATCC 33478), were used in the study. Bacteria were cultured in Brain Heart Infusion (BHI) media and incubated overnight at 37 °C anaerobically in 5% CO_2_ conditions. Heat-killed bacteria were prepared by incubating the culture at 100 °C for 15 min.

### 2.3. Treatment of Candida albicans Biofilm with Heat-Killed Bacteria

To investigate the molecular responses of *C. albicans* biofilms exposed to various oral bacteria, a mature *C. albicans* biofilm (48 h) was treated with a specific, 10^8^ heat-killed bacteria reconstituted in GMM. An aliquot of 200 µL of the suspension was inoculated into wells of a 96-well plate. The cultures were then incubated at 37 °C for 2 h before proteins from the biofilm biomass were extracted for label free mass spectrometry analysis.

### 2.4. Protein Extraction and Sample Preparation for Mass Spectrometry Analysis

Biofilms untreated or treated with bacteria were washed once with PBS and biofilm cells were lysed with yeast protein extraction reagent (Y-PER) (Pierce Biotechnology, Rockford, IL, USA, catalog number: 78991), according to the manufacturer’s protocol. The cell lysates were collected by centrifuging at 18,000× *g* for 10 min and protein concentrations in the supernatants were estimated using Bradford assay (Bio-Rad Laboratories, Hercules, CA, USA, catalog number: 500-0006). An amount of 100 µg of proteins in the solution were then mixed with acetone at a ratio of 1:5 (*v*/*v*) and kept at −20 °C for 30 min. Thereafter, samples were centrifuged at 18,000× *g* for 10 min and supernatants were discarded. Precipitated proteins were dissolved in 8 M urea in 0.1 M Tris-HCl, reduced by 20 mM dithiothreitol, and alkylated with 25 mM iodoacetamide. Following reduction and alkylation, 1 µg of trypsin per 20 µg of proteins were added and trypsinization was performed at 37 °C for 16 h. The digested peptides were then desalted and purified using C18 StageTips prior to LC-MS analysis.

### 2.5. LC-MS Analysis

The liquid chromatography-tandem mass spectrometry (LC-MS/MS) analysis were conducted on a nanoflow high-performance liquid chromatography (HPLC) coupled to an LTQ Orbitrap Velos mass spectrometer (Thermo Fisher Scientific, Waltham, MA, USA). Peptides prepared from previous step were dried down using Savant SpeedVac SPD2010 (Thermo Fisher Scientific, UK) and dissolved in buffer A (99.9% (*v*/*v*) water and 0.1% (*v*/*v*) formic acid). Subsequently, peptide samples were separated by reverse-phase chromatography via an in-house PicoTip column (New Objective, Woburn, MA, USA) (360 μm outer diameter, 75 μm inner diameter, 15 μm tip), packed with 8 to 10 cm of octadecyl-silica (ODS)-A C18 5-μm phase (YMC, Allentown, PA, USA). The flow rate was set at 300 nL/min using linear gradient of mobile B (99.9% (*v*/*v*) acetonitrile and 0.1% (*v*/*v*) formic acid): 0% for 5 min, followed by 2–35% for 150 min. The LTQ Orbitrap was operated in a data-dependent mode through a full scan mode from 350–2000 Da, followed by collision-induced dissociation on the 20 most abundant ions.

### 2.6. Protein Identification

Peptide identifications were carried out on the SearchGUI (version 3.1.3) [16] and Peptideshaker software (version 1.13.6) [17] using the using X!Tandem (version VENGEANCE, 2015.12.15) and the Open Mass Spectrometry Search Algorithm (OMSSA) (version 1.0) with integrated false discovery rate (FDR) analysis function [18]. All peptide identifications were performed in three biological replicates and three technical replicates. The data were searched against a protein sequence database downloaded from UniProtKB for *C. albicans* SC5314 on 20th Oct 2016 (total 9038 entries). The MS/MS spectra obtained were searched using the following user-defined search parameters: Digestion: Trypsin with maximum of 2 missed cleavages; 20.0 ppm as MS1 and 0.6 Da as MS2 tolerances; fixed modifications: carbamidomethyl c (+57.021464 Da); variable modifications: oxidation of m (+15.994915 Da), acetylation of protein *n*-term (+42.010565 Da), pyro-cmc (−17.026549 Da), pyro-glu from *n*-term e (−18.010565 Da), and pyro-glu from *n*-term q (−17.026549 Da). The MS/MS spectra were searched against a decoy database to estimate the false discovery rate (FDR) for peptide identification. The decoy database consisted of reversed protein sequences from the *C. albicans* database. Peptides and proteins were inferred from the spectrum identification results using PeptideShaker. Peptide spectrum matches, peptides, and proteins were validated at a 1.0% false discovery rate estimated using the decoy hit distribution.

### 2.7. Protein Quantification from Spectral Counts

Differences in spectral counts are identified by applying a likelihood ratio test (G test) for independence as described previously [19]. *p*-value of less than 0.05 in the G statistic was considered significant. To quantify changes in spectral counts, we estimated fold changes as proposed by Beissbarth et al. [20], for serial analysis of gene expression (SAGE) data as follow:R_SC_ = log_2_[(n_2_ + f)/(n_1_ + f)] + log_2_[(t_1_ − n_1_ + f)/(t_2_ − n_2_ + f)](1)
where, for each protein, relative spectral counts (R_SC_) is a log_2_ ratio of abundance between 2 conditions; n_1_ and n_2_ are spectral counts for the proteins in two conditions; t_1_ and t_2_ are the total number of spectra over all proteins in the two conditions; and f is a correction factor set to 0.5 by Beissbarth et al. [20].

### 2.8. Differential Protein Expression Analysis and Cluster Analysis

Differences between *Candida albicans* SC5314 biofilms’ protein expression profiling when exposed to oral bacteria were analyzed using limma package [12] from Bioconductor as previously described [21]. Cluster analysis [13] was also used to arrange proteins according to similarity in pattern of expression.

### 2.9. Pathway Analysis

Identified proteins were subjected to gene ontology analysis using Cytoscape (v3.4.0) [22] with BINGO plugin (v3.0.3) [23] as previously described [15,21,24].

## 3. Results

### 3.1. Proteomic Analysis of Candida albicans Biofilms Identified Proteins Expressed on Exposure to Heat-Killed Bacteria

Using a decoy search strategy [25], false discovery rate (FDR) for protein and peptide were estimated to be <1%. Differences in spectral counts between *C. albicans* SC5314 biofilms untreated (control) and treated with heat-killed bacteria were used to estimate changes in protein fold expression with normalization to total spectral counts to account for loading differences. Figure 1 illustrates the volcano plots analyzing the significance patterns of protein expressed in *C. albicans* biofilms for each bacterial exposure. By overlapping the differentially expressed proteins, 164 proteins were identified to be consistently lower, and three proteins to be consistently higher in bacteria-treated *C. albicans* biofilms (Appendix A). Functional annotation of these proteins using Cytoscape with BINGO plugin, revealed that the proteins related to cellular respiration, translation, oxidoreductase activity and ligase activity were altered when exposed to the chosen panel of the heat-killed oral bacteria (Figure 2).

### 3.2. Differential Expression Analysis Uncovered Proteomic Features of Candida albicans Biofilms Responding to Gram-Positive and -Negative Bacteria

Aforementioned analysis unraveled *C. albicans* proteins were variably expressed when exposed to different bacterial strains. Therefore, in the next analysis, *C. albicans* proteomic responses were classified according to their Gram staining characteristics, a surrogate indicator of the cell wall structure as illustrated in Figure 3A. Subsequent differential expression analysis revealed a total of 35 significantly altered proteins (Appendix A). Pathway analysis of these proteins suggested that the enzyme regulator activity and ligase activity were significantly lower when *C. albicans* were exposed to Gram-negative bacteria (Figure 3B), while the proteomes related to cellular respiration and cytoplasmic proteins were significantly lower on exposure to Gram-positive bacteria (Figure 3C).

### 3.3. Cluster Analysis Revealed C. albicans Biofilm Proteins with Similar Expression Profiles on Bacterial Exposure

We further exploited cluster analysis to classify the proteomic changes according to the similarity in expression profiling [13]. Interestingly, cluster analysis divided the *Candida albicans* responses into four different treatment groups: (1) *A. naeslundii*, *E. faecalis*, and *P. gingivalis*; (2) *A. actinomycetemcomitans* and *F. nucleatum*; (3) *S. mutans* and *S. sanguinis;* and (4) *S. mitis* and *S. sobrinus* (Figure 4). Subsequent differential and pathway analysis identified proteins that were upregulated (Figure 5) or downregulated (Figure 6) in *Candida albicans* biofilm proteomics responses as per the different treatment groups (*p*-value < 0.05).

*C. albicans* biofilms treated with group 1 bacteria, *A. naeslundii*, *E. faecalis*, and *P. gingivalis*, displayed higher expression levels of proteins related to ribosome and translation process (Figure 5A). In contrast, enzymatic activities (lyase activity, ligase activity, isomerase activity and transferase activity), energy generation, carbohydrate metabolism, protein folding, and cell wall protein expression were simultaneously downregulated in *C. albicans* biofilms when exposed to group 1 bacteria (Figure 6A).

*C. albicans* biofilms treated with group 2 bacteria, *A. actinomycetemcomitans* and *F. nucleatum*, exhibited reduced levels of protein expression related to mitochondrial and oxidoreductase activity (Figure 6B). As differential expression analysis in our previous study did not reveal upregulation of proteins on exposure to group 2 bacteria, gene ontology analysis was not performed for this component of the study.

On the other hand, *C. albicans* biofilms treated with group 3 bacteria, *S. mutans* and *S. sanguinis*, displayed higher expression levels of proteins related to ligase activity, polarized growth, and plasma membrane synthesis (Figure 5B). Conversely, proteins related to ribosome, translation process, and structural molecule activity were significantly downregulated (Figure 6C).

Lastly, *C. albicans* biofilms treated with group 4 bacteria, *S. mitis* and *S. sobrinus*, expressed significant quantities of proteins related to cell wall synthesis (Figure 5C and Figure 6D). Proteins related to ribosome, translation process, and enzymatic activities (ligase activity and lyase activity) were upregulated (Figure 5C). Whereas proteins related to cellular homeostasis, carbohydrate metabolism, and energy generation were downregulated (Figure 6D).

Interestingly, by correlating the cell sizes of the bacterial species, we were able to discern the emergence of a consistent cluster profile. For instance, a cluster of groups 3 and 4 bacteria (the *Streptococcus* groups) with the smallest cell size (of around 0.5 to 0.75 µm), followed by a cluster comprising group 1 bacteria (*A. naeslundii, E. faecalis*, and *P. gingivalis*) having a marginally larger cell size (0.5 to 10 µm), and finally, the group 2 bacterial cluster with the largest and varied cell dimensions (5 to 50 µm) (Table 1). Therefore, we hypothesized that bacteria cell dimensions could be a determining factor governing *C. albicans* proteome response.

## 4. Discussion

Here, we report using differential expression analysis and cluster analysis the varied proteome expression pathways involved in bacterial-*C. albicans* interactions in biofilms. Our study also suggests that the bacterial Gram staining characteristics, the species, their cell size, and morphologies play an important role in determining the proteomic responses associated with the mutualistic interactions in candidal-bacterial biofilms.

Exposure to bacteria induced changes in cellular respiration profile of *C. albicans* biofilms (Figure 2 and Appendix A), suggesting an effective metabolic adaptation taking place to enhance its survival and pathogenesis. Similar observations in *Pseudomonas-C. albicans* interkingdom interactions have been reported by Morales et al. [26], thus corroborating our findings. The latter group reported that diminishing respiration had a survival value in an ecosystem within a biofilm that competes for limited resources such as oxygen and food [26]. Interestingly, fluctuations in cellular respiration of *C. albicans* are reported to be associated with morphogenesis of the fungal cells associated with yeast to hyphal transition [27]. Hence the reduction in cellular respiration we noted in *C. albicans* biofilms on exposure to bacteria may have endurance value for the pathogen both in terms of resource utilization, and morphogenesis.

Differential proteomic analysis revealed that the significant differences of *C. albicans* biofilms upon exposure to Gram-positive and Gram-negative bacteria. Gram-positive and -negative bacteria are classified due to their differences in cell wall components; Gram-positive bacteria contain a thick polymer of amino acids and sugars layer of peptidoglycans (PGN) in the cell wall, which takes up to 90% of their dry weight [28]. On the other hand, Gram-negative bacteria have a thinner layer of PGN, making up only 10% of their dry weight, but they are characterized by a cell wall outer membrane of lipopolysaccharides (LPS) that gives them their distinctive properties (Figure 3A). It is therefore tempting to speculate that the different cell wall components of Gram-negative and -positive bacteria may play a role in the differential proteomic expression we noted in *Candida* biofilms. Consistent with our findings, these chemical structures have been reported to incite molecular changes in *C. albicans* [29,30,31,32]. Both LPS and teichoic acid have been shown to reduce adhesion of *C. albicans* to host cells [29,30,31]; while PGN is reported to regulate *C. albicans* virulence via activation of Cyr1 protein [32]. Specifically, muramyl-l-alanine-d-isoglutamine (MDP, a peptidoglycan subunit) has been identified for activating the important Ras1-Cyr1 signaling pathway [33,34].

On the other hand, on cluster analysis, we noted differential and distinct protein expression profiles in the *C. albicans* biofilm proteome exposed to different groups of oral bacteria (Figure 4, Figure 5 and Figure 6). Interactions between *C. albicans* biofilms and heat-killed *A. naeslundii*, *E. faecalis*, and *P. gingivalis* seemed to reduce expression of cell wall proteins and the carbohydrate metabolism in the biofilms (Figure 6A). Other studies, which substantiates our findings, have also reported that *A. naeslundii, E. faecalis*, and *P. gingivalis* induce an antagonistic effect on *C. albicans* growth and enzymatic metabolism [35,36,37]. Our study also revealed that exposing *C. albicans* biofilms to *A. actinomycetemcomitans* or *F. nucleatum* decreases the mitochondrial and oxidoreductase activity of the biofilms (Figure 6B). Mitochondria functions have been implied to play a role in *C. albicans* antifungal responses and hyphal formation [27]. As *A. actinomycetemcomitans* and *F. nucleatum* have been shown to inhibit hyphal formation in *C. albicans* [38,39], our data imply that *A. actinomycetemcomitans* and *F. nucleatum* induce changes in mitochondrial activity of *C. albicans*, which may subsequently lead to hyphal inhibition. Lastly, our analysis also identified significant protein alternations in plasma membrane, polarized growth, and cell walls of *C. albicans* biofilms as a response to streptococcal stimulation. Others too have uncovered the synergistic activity between *Streptococcus* species and *C. albicans* [40,41,42,43], one of which alludes to the augmentation of *C. albicans* biofilm development by *S. mutans* enzyme, GtfB [43]. Our study, therefore, has provided further substantiation of *Streptococcus-C. albicans* interactions, with a particular focus on the molecular features of *Candida* stimulated by cellular structures of the heat-killed streptococci.

Finally, for the cluster analysis of the differential protein expression by the yeast due to bacterial exposure we noted an association between bacterial sizes and morphologies and the resultant *C. albicans* molecular responses (Figure 4 and Table 1). Our data tend to imply that smaller cell sized bacteria induced changes in expression of cell wall proteins of the yeast, while the larger cell sized bacteria lowered the expression of proteins related to carbohydrate metabolism and mitochondria components. Therefore, it is hypothesized that larger bacteria are likely to display antagonistic effects on *C. albicans* biofilms as compared with the smaller bacteria.

In conclusion, this proteomic study has, for the first time, demonstrated, through differential and cluster analysis, the intricate yet subtle changes in the proteome of *Candida albicans* in mixed bacterial-fungal, interkingdom biofilms, that are particularly common in the human oral eco-niche. In particular, the proteome of the fungus associated with cellular respiration, translation, oxidoreductase activity and ligase activity were significantly altered on exposure to the cohabitant bacteria. Additionally, the *C. albicans* proteome appeared to be impacted by particular bacterial species, their cell morphologies, and Gram staining characteristics. However, further work is essential to confirm the current data, and to demystify the intricate chemical operative mechanisms underlying fungal–bacterial interactions within biofilms.

## Figures and Tables

**Figure 1 microorganisms-08-01541-f001:**
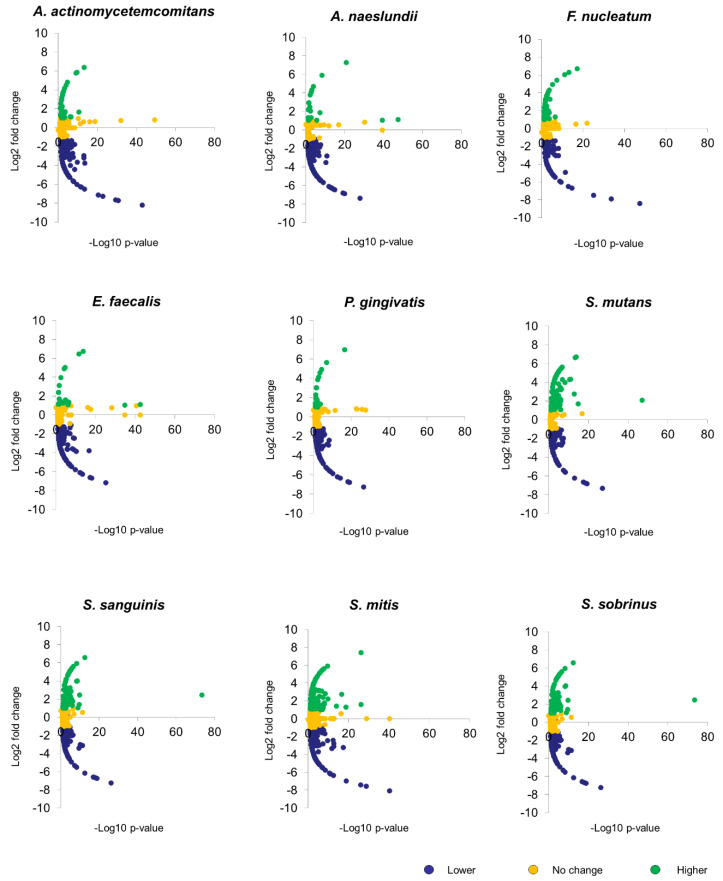
Volcano plots of protein expression changes of *C. albicans* biofilms. Biofilm development of *C. albicans* SC5314 was conducted at 37 °C. The 48 h mature biofilms were treated with 10^8^ heat-killed bacteria for 2 h at 37 °C. Proteins were then extracted and sent for label free mass spectrometry analysis. Volcano plots illustrated significance versus fold changes of different proteins expressed in *C. albicans* biofilms under stimulations of *A. actinomycetemcomitans, A. naeslundii, F. nucleatum, E. faecalis, P. gingivalis, S. mutans, S. sanguinis, S. mitis*, and *S. sobrinus.*

**Figure 2 microorganisms-08-01541-f002:**
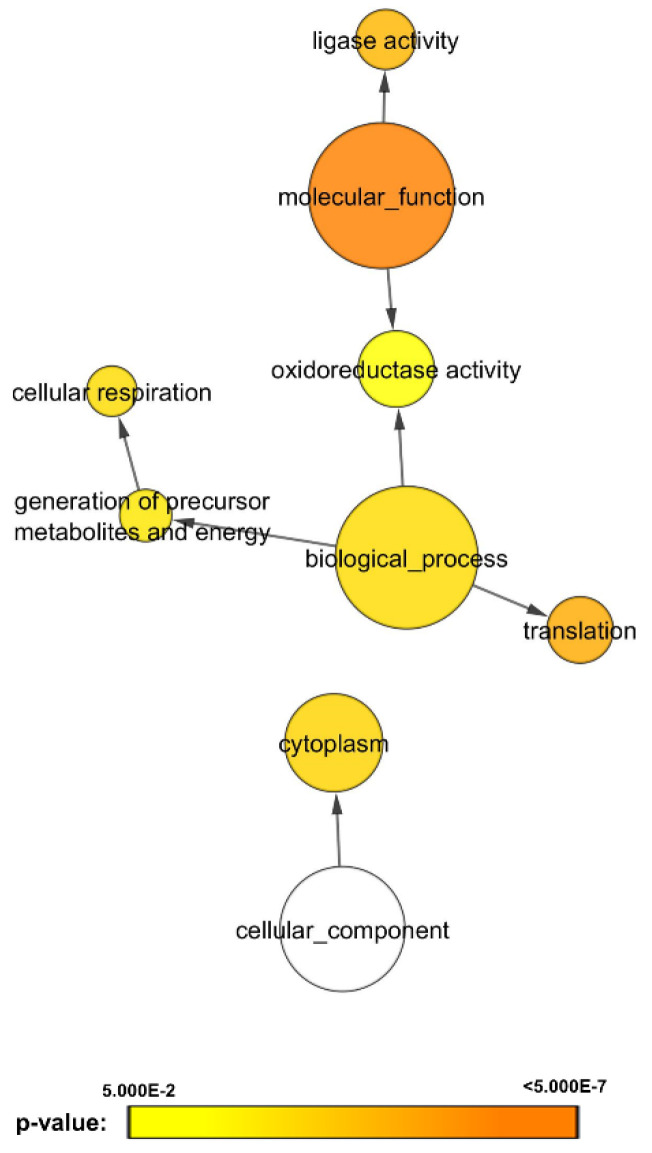
Gene ontology analysis of proteins commonly expressed in *C. albicans* biofilms. Analysis of overlapping differentially expressed proteins under bacterial stimulations revealed 164 proteins to be consistently downregulated and three proteins to be consistently upregulated. Biological processes, molecular functions, and cellular components were found to be significantly altered in *C. albicans* biofilms under stimulations of nine oral bacteria species; *A. actinomycetemcomitans, A. naeslundii, F. nucleatum, E. faecalis, P. gingivalis, S. mutans, S. sanguinis, S. mitis*, and *S. sobrinus* bacteria (*p*-value < 0.05). The protein expression levels were represented in *p*-value using different colors: yellow (least significant) to orange (most significant). The white circle represents cellular component, a distinct aspect of how gene functions can be described.

**Figure 3 microorganisms-08-01541-f003:**
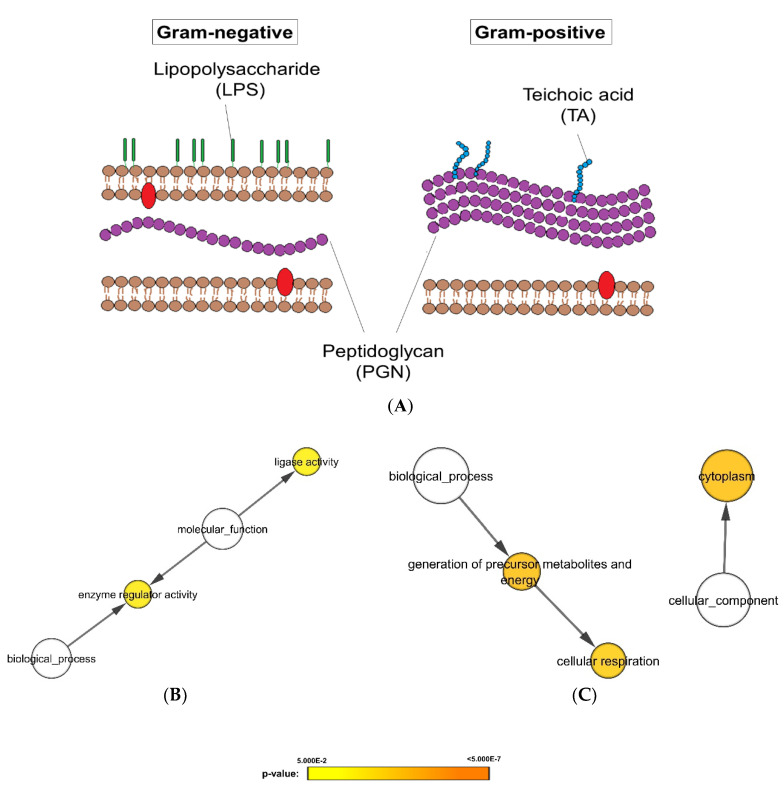
Differential expression and gene ontology analyses of proteomes of *C. albicans* biofilms when exposed to Gram-negative and Gram-positive heat-killed bacteria. (**A**) Schematic illustration of differences in cell wall components between Gram-negative and Gram-positive bacteria. Biological processes, molecular functions, and cellular components that are significantly lower in *Candida* biofilm exposed to (**B**) Gram-negative bacteria or (**C**) Gram-positive bacteria (*p*-value < 0.05). The protein expression levels were represented in *p*-value using different colors: yellow (least significant) to orange (most significant). The white circles describe the 3 distinct aspects of how gene functions can be described: molecular function, cellular component, and biological process.

**Figure 4 microorganisms-08-01541-f004:**
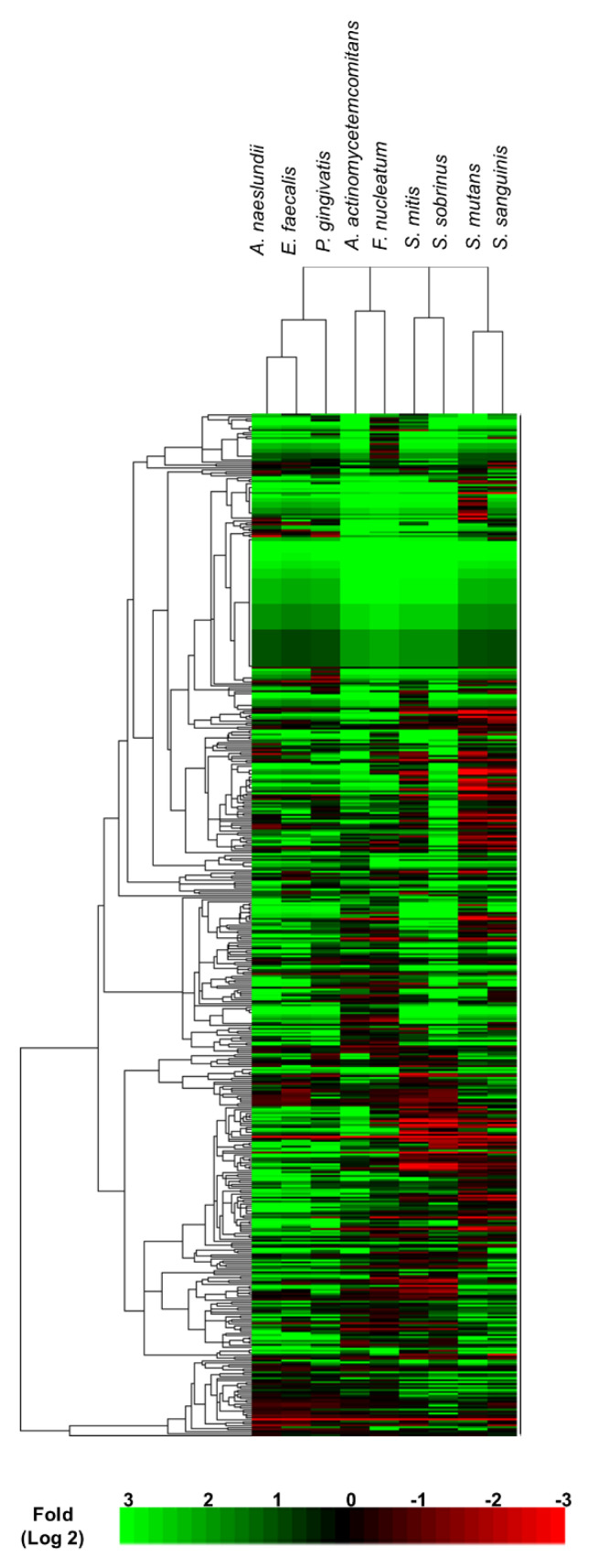
Cluster analysis of proteomic responses of *C. albicans* biofilms under bacterial exposures. Cluster software was used to estimate Euclidean distances between the proteomes and TreeView was used to illustrate the proteome profiles and linkages (*p*-value < 0.05).

**Figure 5 microorganisms-08-01541-f005:**
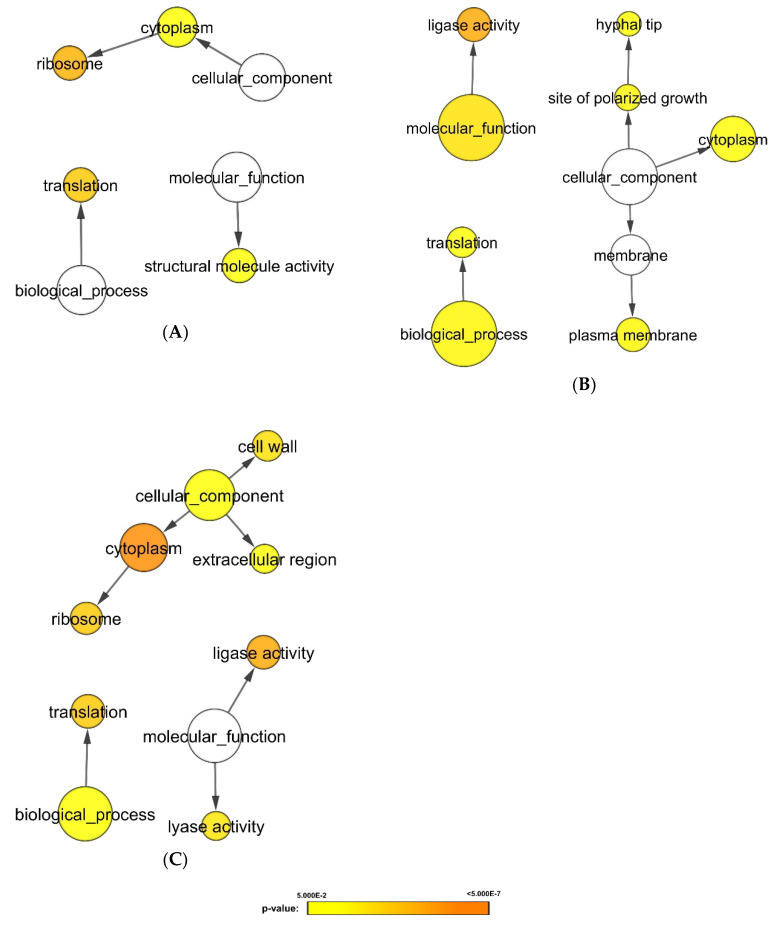
Gene ontology analysis of proteins with higher expression in *C. albicans* biofilms when exposed to different groups of bacteria. Proteins extracted from 48 h mature biofilms were sent for label free mass spectrometry analysis. Differential expression analysis was first applied to determined biofilms’ proteins which were higher in each cluster. Subsequent gene ontology analysis illustrated biological processes which were significant higher in the *C. albicans* under bacterial stimulants: (**A**) Group 1: *A. naeslundii*, *E. faecalis*, and *P. gingivalis*; (**B**) group 3: *S. mutans* and *S. sanguinis*; (**C**) group 4: *S. mitis* and *S. sobrinus* (*p*-value < 0.05). The protein expression levels were represented in *p*-value using different colors: yellow (least significant) to orange (most significant). The white circles describe the 3 distinct aspects of how gene functions can be described: molecular function, cellular component, and biological process.

**Figure 6 microorganisms-08-01541-f006:**
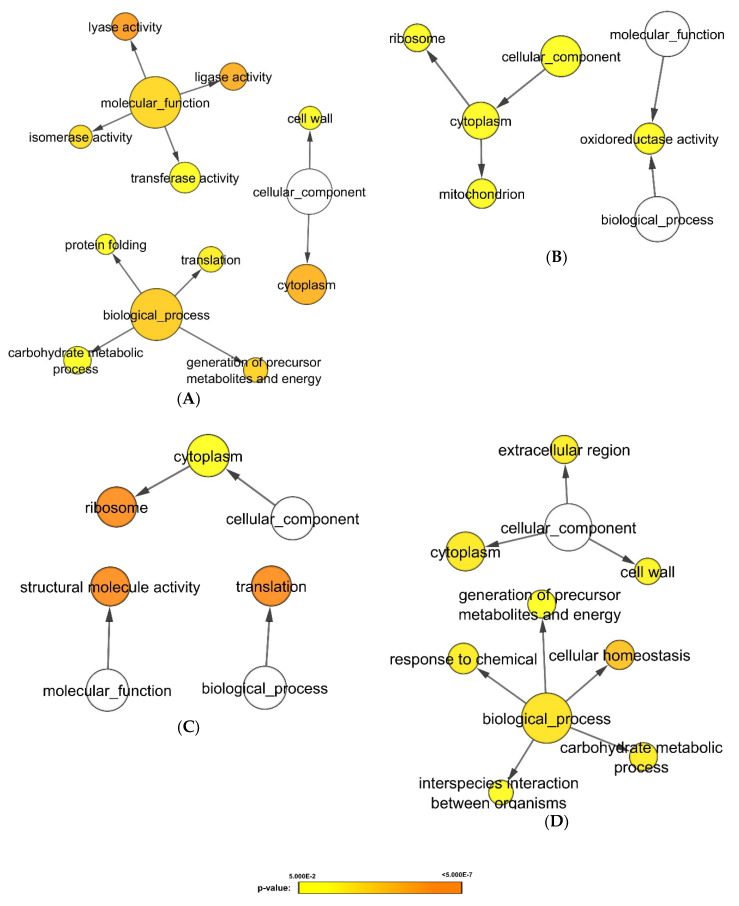
Gene ontology analysis of proteins with lower expression in *C. albicans* biofilms when exposed to different groups of bacteria. Proteins extracted from 48 h mature biofilms were sent for label free mass spectrometry analysis. Differential expression analysis was first applied to determine biofilms’ proteins which were lower in each cluster. Subsequent gene ontology analysis illustrated biological processes which were significant lower in the *C. albicans* under bacterial stimulants: (**A**) Group 1: *A. naeslundii*, *E. faecalis*, and *P. gingivalis*; (**B**) group 2: *A. actinomycetemcomitans* and *F. nucleatum*; (**C**) group 3: *S. mutans* and *S. sanguinis*; (**D**) group 4: *S. mitis* and *S. sobrinus* (*p*-value < 0.05). The protein expression levels were represented in *p*-value using different colors: yellow (least significant) to orange (most significant). The white circles describe the 3 distinct aspects of how gene functions can be described: molecular function, cellular component, and biological process.

**Table 1 microorganisms-08-01541-t001:** Sizes and morphologies of tested bacterial species.

Bacterial Species	Group	Gram	Size	Morphology
*E. faecalis*	1	Positive	0.6–2.5 µm	Cocci (spherical or ovoid)
*P. gingivalis*	1	Negative	0.5–5.0 µm in length	Bacillus
*A. naeslundii*	1	Positive	5–10 µm in length	Bacillus
*F. nucleatum*	2	Negative	5–50 µm in length, varied	Bacillus
*A. actinomycetemcomitans*	2	Negative	0.2–1 µm in diameter, varied	Coccobacillus
*S. mutans*	3	Positive	0.5–0.75 µm	Cocci (spherical)
*S. sanguinis*	3	Positive	0.5–0.75 µm	Cocci (spherical)
*S. mitis*	4	Positive	0.5–0.75 µm	Cocci (spherical)
*S. sobrinus*	4	Positive	0.5–0.75 µm	Cocci (spherical)

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
