# Peer review of "The Proteome of Community Living Candida albicans Is Differentially Modulated by the Morphologic and Structural Features of the Bacterial Cohabitants"

_microorganisms, 2020, doi:10.3390/microorganisms8101541_

Round 1
Reviewer 1 Report
The article "The proteome of community living Candida albicans is differentially modulated by the morphologic and structural features of the bacterial cohabitants" by T. Truong et al. presented for review, concerns changes in the proteome of the biofilm formed by C. Albicans yeast in contact with selected bacteria. The work is at a high scientific and substantive level and constitutes a significant contribution to the development of the field. The chapter "introduction" presents the background of the research in a concise and legible way. The used research methods are adequate for the research problem and allow for its solution. The authors consciously use research tools and the obtained results are presented legibly and correctly interpreted.
Despite the high scientific level of the work, the reviewer raises some questions and ambiguities that I would ask the authors to explain.
General thoughts:
1. The subject of the research is the evaluation of the effect of selected bacteria on the change of the proteome of the biofilm formed by C. Albicans. Why were heat-killed bacteria used in the research but not living microorganisms?
2. The use of dead bacteria is not relevant to the physiological condition in which live yeasts and bacteria coexist. Live bacteria release a number of yeast modulating factors as well as enzymes that significantly alter yeast response. Have the authors attempted to analyze yeast proteomic changes under the influence of live bacteria?
Introduction:
3. In lines 66-68 the authors list the bacteria used. The reviewer suggests dividing bacteria into gram-positive and gram-negative at this point, especially that the authors refer to this criterion in the following sections. The only place where this information appears is in Table 1.
4. Lines 75-76 show that bacteria modulate yeast proteome. The conducted research uses killed bacteria, so the statement that bacteria modulate the proteome is somewhat of an overinterpretation. It would be appropriate to define it as heat-killed bacteria or cell walls.
Materials and methods:
5. Line 84 (96 well plate) - I suggest writing down the catalog number of the plates. Were "high binding" plates used? Standard plates do not provide an adequate surface for yeast adhesion.
6. Line 87 - why was the medium changed after 24h? Removal of the medium also changes the environment, along with the removal of factors released into the environment that modulate biofilm formation, including Quorum Sensing Molecules, ie AI-2.
7. Line 86-87 - C. Albicans growth time - why the analysis was carried out after 48 hours? The growth of yeast is divided into three stages (10.17305/bjbms.2017.1667): early (0-3h), invasion (3-12h), late (12-24h). The use of yeast after 48h simulates the emergence of co-infection relatively late when the yeast proteome is stable. Did the authors carry out analogous analyzes for yeast after e.g. 18-24 hours of development?
8. Lines 86-97 - plates were incubated at 37 °C. The authors do not specify what morphological form of yeast was used in the study. The methodology suggests a filamentous form, but was this form obtained in the GMM medium? For the induction of the filamentous form, supplementation with 10% serum (10.1242/jcs.002931) or 5% FBS (10.3390/microorganisms8010075) is necessary. I am asking for a correction of this part of the methodology. Were microscopic images of the obtained filamentous form of yeast taken? It is worth supplementing the manuscript with such pictures.
9. Lines 90-93 - Bacterial culture conditions - were the bacteria grown aerobically or anaerobically? The anaerobes are among the selected microorganisms, while the authors do not mention the culture conditions. Please improve this part of the methodology.
10. Lines 93-94 - the bacteria were killed at 100 °C for 15 minutes. Was the viability of the bacteria verified after this procedure? For example, MTT or measurement of change in OD over time in the medium.
11. Lines 99-101 - why the analysis of proteome changes was carried out after 2 hours of contact with heat-killed bacteria? Were other co-incubation times also checked? This time may be important for changes in the proteome (10.17305/bjbms.2017.1667). Similarly, the relationship between the stage of yeast development vs. co-incubation time may be important.
Results:
12. Lines 175-181 - a repetition of the materials and methods chapter, I suggest removing it.
Discussion:
13. Could the authors try to pinpoint a potential bacterial size signaling mechanism or molecules affecting the yeast proteome?
Author Response
We thank the reviewer for the expert opinions and suggestions. We have provided a point-by-point reply to your comments. Please see the attachment.

Reviewer 2 Report
The oral cavity comprises the most complex niches of the human body colonized by a wide variety of fungi and bacteria. These commensally or often opportunistic and pathogenic colonizers tend to form biofilms, which are directly attributable to the virulence of these microorganisms and their ability to cause infections. Therefore, the study of the complexes interactions of these biofilm is very important.
The present paper proposes the proteomic interactions of C. albicans biofilms with different common oral bacterial species using limma differential expression analysis, and cluster analysis. Very interesting idea, interesting preliminary results, I hope the author will continue the research in this direction.
The article is written in a concise manner and is relatively easy to follow. However, I would suggest a few small additions.
On the material and method part, the identification data of the bacterial strains are missing. Are they ATCC strains? Please specify. I think it would be worth mentioning the source of both the microbial strains and the reagents used in the study.
I also found it hard to understand Figures 5 and 6. The white, gray or dark gray circles are quite confusing and have no legend. Maybe the authors find a way to complete or modify them to make it easier to understand these figures.
Author Response
We thank the reviewer for taking time to express concerns and suggestions to improve on the current manuscript. We have taken reviewer’s comments into consideration to revise the manuscript accordingly. Please see the attachment.
